# Birth month and infant gross motor development: Results from the Japan Environment and Children's Study (JECS)

**Kahoko Yasumitsu-Lovell**[1,2], **Lucy Thompson**[1,3], **Elisabeth Fernell**[1], **Masamitsu Eitoku**[2], **Narufumi Suganuma**[2]*, **Christopher Gillberg**[1,3], **the Japan Environment and Children's Study Group**[¶]

**1** Gillberg Neuropsychiatry Centre, Sahlgrenska Academy, University of Gothenburg, Gothenburg, Sweden, **2** Department of Environmental Medicine, Kochi Medical School, Kochi University, Nankoku, Kochi, Japan, **3** Institute of Health and Wellbeing, University of Glasgow, Glasgow, Scotland

¶ Membership of the Japan Environment and Children's Study Group is provided in the Acknowledgments
* nsuganuma@kochi-u.ac.jp

**Data Availability Statement:** Data are unsuitable for public deposition due to ethical restrictions and legal framework of Japan. It is prohibited by the Act

## Abstract

The association between birth month and neurodevelopmental or psychiatric disorders has been investigated in a number of previous studies; however, the results have been inconsistent. This study investigated the association between birth month and child gross motor development at 6 and 12 months of age in a large cohort of infants (n = 72,203) participating in the Japan Environment and Children's Study (JECS). Gross motor development was assessed using the Ages and Stages Questionnaire (ASQ-3). At 6 months and 12 months, 20.7% and 14.2%, respectively, had ASQ-3 indications of gross motor problems. Birth month was strongly associated with gross motor development at both time points, particularly at 6 months. Summer-born infants had the worst outcomes at both 6 months and 12 months of age. This outcome applied to the ASQ-3 score itself and to the adjusted Relative Risk (aRR), with the highest aRRs (relative to January-born) among August-born (aRR 2.51; 95%CI 2.27–2.78 at 6 months), and June-born (aRR 1.84; 95%CI 1.63–2.09 at 12 months). Boys had better scores than girls both at 6 and 12 months of age. We speculate that seasonal factors—such as maternal vitamin D deficiency and influenza infection—affecting the fetus in early pregnancy might account for the findings.

## Introduction

Birth month is one of the factors affecting lifetime disease risk, and distinct patterns have been observed across various diseases including cardiovascular, respiratory, reproductive, and neurological problems [1]. A number of studies have investigated the effects of birth month or season on neurological, neuropsychiatric, and neurodevelopmental disorders, such as epilepsy, autism spectrum disorder (ASD), attention-deficit/hyperactivity disorder (ADHD), and learning disorders (LDs) (Procopio et al 2006, Gillberg 1990, Landau et al 1999, Boland et al 2015, Mackay et al 2016), as well as on schizophrenia and depression (Gadow and DeVincent 2012,

on the Protection of Personal Information (Act No. 57 of 30 May 2003, amendment on 9 September 2015) to publicly deposit the data containing personal information. Ethical Guidelines for Medical and Health Research Involving Human Subjects enforced by the Japan Ministry of Education, Culture, Sports, Science and Technology and the Ministry of Health, Labour and Welfare also restricts the open sharing of the epidemiologic data. All inquiries about access to data should be sent to: jecs-en@nies.go.jp. The person responsible for handling enquiries sent to this e-mail address is Dr Shoji F. Nakayama, JECS Programme Office, National Institute for Environmental Studies. URL https://www.env.go.jp/chemi/ceh/en/index.html. The authors had no special access privileges to the data others would not have.

**Funding:** The Japan Environment and Children's Study is funded by the Ministry of the Environment, Japan.

**Competing interests:** The authors have declared that no competing interests exist.

Disanto et al 2012) [1–8]. However, the results have been inconsistent. For instance, birth seasons/months with the highest risk of ASD are heterogeneous, ranging from spring [3,9], to summer/autumn [10,11], or no association [11]. This inconsistency most likely derives from methodological differences, such as study designs, study regions, and sample size.

Early abnormal or unusual neuromotor development in a child, particularly gross motor development, may indicate a later-appearing neurodevelopmental disorder, regarded as the "dominant domain of neurodevelopment, particularly the first year of life [12]" and a possible early marker for ASD [12,13]. However, like the discrepant findings of birth season effects on neurodevelopmental disorders, the results of a limited number of previous studies on the association between birth month/season and early childhood motor development have also varied across different countries in the northern hemisphere, ranging from winter/spring-born outperforming others to no association [8,12,14–16]. Tsuchiya et al. [12] concluded that among 742 Japanese infants, although the winter/spring-born were more advanced in their motor development at 6 and 10 months of age, the advantage disappeared at 14 months; in contrast, McGrath et al. [8] found that among 22,123 American children, the winter/spring-born showed better motor development all the way from 8 months up to 7 years of age. These discrepancies most likely derive from the small sample size of the studies—less than several hundred participants, except the study by McGrath et al.—as well as variation in country, climates, screening tools, and timing of outcome measurements. Therefore, a large-scale cohort study on the association between birth month and early gross motor development is warranted in order to elucidate the association between birth season and gross motor development.

The aim of the present study was to investigate the association between birth month and gross motor development at 6 and 12 months, utilizing the data from one of the world's largest (100,000 participants) ongoing nationwide birth cohort studies, the Japan Environment and Children's Study (JECS). As Japan is geographically extended and climate types vary from subarctic to subtropical, latitude is also considered as a possible proxy for other seasonal and biometeorological factors.

## Methods

The present investigation is a prospective study within the JECS, where the main goal is to identify environmental factors that affect children's health and development (Dataset: jecs-an-20180131). A total of 104,065 fetal records were registered between January 2011 and March 2014 throughout Japan, with 100,144 live births. To ensure the generalizability, fifteen Regional Centres were selected from the northern (Hokkaido) to southern (Okinawa) ends of Japan, covering all four major islands. Participants were recruited at cooperating health care providers and/or local government offices where pregnant women register themselves, with a targeted coverage rate more than 50% in the study areas [17]. The inclusion criteria were as follows: *1*) residents of the study areas at the time of recruitment; *2*) due date between 1 August 2011 and mid-2014; and *3*) sufficient comprehension of the Japanese language [17]. The baseline characteristics of the JECS mother-child dyads were comparable with those obtained in the national survey in 2013 [18]. Detailed information on the JECS protocol and its representativeness can be found elsewhere [17,18].

The data utilized in the present study comprised a part of the JECS data, collected between the enrollment during pregnancy and the first year of life, from self-administered questionnaires (twice during pregnancy, and when the child was 1, 6, and 12 months of age) and a review of medical records. The questionnaires covered various topics, including lifestyle, socio-economic status (SES), diet, medical history, and medication [19]. The medical record

transcripts were obtained directly from the hospitals/clinics at enrollment, birth, and the child's one-month-old examination.

The eligible participants for this study were full-term births (37 or more gestational weeks) with a complete set of information on birth month (in the medical record transcripts at birth), maternal vitamin D intake (calculated from the Food Frequency Questionnaire during pregnancy), and the Ages and Stages Questionnaire (ASQ-3) scores at 6 and 12 months of age. The final sample was 72,203, 72% of the live births of the JECS (n = 100,144).

The exposure variable was birth month. As birth month is a possible proxy for other seasonal factors, geographical and meteorological data (latitude, monthly UV index, pressure, precipitation, temperature, and sunlight duration) pertaining to the location of the JECS Regional Centres were merged with the JECS dataset [20].

Outcomes were measured by parents or guardians with the ASQ-3 when the child was 6 and 12 months old. The ASQ-3 consists of five skill domains: communication, gross motor, fine motor, problem solving, and personal-social. Each domain contains six questions, with the answers scored as follows: Yes = 10, Sometimes = 5, Not yet = 0. Thus, the maximum positive score in each domain was 60. The internal consistency of the ASQ-3 for the present sample was assessed with Cronbach's alpha (0.76 at 6 months and 0.78 at 12 months). The present study focused particularly on the gross motor domain, as gross motor development is a reliable early marker to assess child neurodevelopment [21]. As the Japanese version of the ASQ-3 is currently in the process of validation, the recommended cut-off values for the validated English version were applied, after thorough comparisons with the JECS pilot study results for the Japanese version [22]. The cut-off values for the gross motor domain were 22.25 and 21.49 at 6 and 12 months, respectively, with lower scores indicating need for further professional assessment.

## Statistical analysis

The study participants were born between July 2011 and December 2014. After confirming the absence of any significant differences during these 4 years (2011–2014) were regarding outcome (i.e., ASQ-3 gross motor scores), participants' background, and meteorological agency information, the data were collapsed across birth years. Furthermore, we included latitude as the only representative variable because meteorological factors of UV index, air pressure, precipitation, temperature, sunlight duration were collinear.

The association between gross motor development at each time point (6 and 12 months of age) and birth month was investigated by using modified Poisson regression analysis with a robust variance estimator (with January as the reference) [23–25]. The association between the ASQ-3 gross motor scores and covariates (i.e., latitude; vitamin D intake from food and supplements as a possible beneficial nutrient; frequency of sunscreen usage; daily length of time spent outside; putative pre-, peri-, and post-natal risk factors for neurodevelopment offspring; and socio-economic status (SES)) was also assessed by using the chi-square test and the two-tailed $t$-test. Covariates with significant $p$ values at any time point are described in S1 Table, and included in the final model to calculate the adjusted Relative Risk (aRR) in modified Poisson regression analysis with a robust variance estimator. Multiple imputation by chained equation was conducted to confirm that the results of the final model were reliable even though 23,398 participants (32.5%) were not included in the final adjusted model.

After calculating the aRR at each time point, the participants were divided into the following four groups by the outcome results at 6 and 12 months: BEST (passed the cut-off for the ASQ-3 gross motor domain at both 6 and 12 months), IMPROVED (failed at 6 months but passed at 12 months), WORSENED (passed at 6 months but failed at 12 months), and

WORST (failed at both time points). Finally, we stratified the cohort by child gender to investigate gender differences. All analyses were conducted using Stata Ver.15.0. The significance level was set at $p < 0.05$ with a 95% confidence interval (CI).

The JECS was approved by the institutional review boards (IRBs), in compliance with the Ethical Guidelines for Epidemiological Research (the Ministry of Education, Culture, Sports, Science and Technology (MEXT) and the Ministry of Health, Labour and Welfare (MHLW)), as well as the Ethical Guidelines for Analytical Research on the Human Genome/Genes (MEXT, MHLW and the Ministry of Economics, Trade and Industry). Written informed consent was obtained from all participants.

## Results

A total of 72,203 children (72.1% live births; 36,784 boys and 35,419 girls) were analyzed in the present study (Fig 1). No differences in characteristics between those included and excluded were found. For example, the mean maternal age at enrollment for those included in the present sample was 30.8 (95% CI: 30.0, 30.9) years and 30.3 (95% CI: 30.2, 30.3) years for those excluded. At 6 and 12 months, 14,960 (20.7%) and 10,260 (14.2%) infants, respectively, scored below the cut-off for the ASQ-3 gross motor domain. At 6 months, August- and September-born infants showed the highest percentages (32.0% and 31.1%, respectively) of scoring below the cut-off, whereas February- and March-born infants showed the lowest percentages (9.6% each) (Table 1). At 12 months, the infants with the highest percentages for scoring below the defined cut-off shifted earlier to the June- and July-born (18.4% each), as did those with the lowest percentages to the December-, January-, and February-born (11.6%, 10.6%, and 11.6%, respectively).

Table 2 shows the crude and adjusted RRs of birth month and latitude for gross motor development at 6 and 12 months of age. All aRR values were similar to the crude RR values even after adjustment, with a cyclical trend of being highest for the August- and September-born and lowest for the February- and March-born at 6 months. The aRR at 12 months showed a similar cyclical trend, being highest for the June- and July-born and the lowest for the February-born. In addition, another difference was observed between the outcomes at 6- and 12- months: with January as the reference, the aRRs for the February-, March-, and April-born (0.75, 0.76, and 0.82, respectively; $p<0.05$) implied that these birth months were "protective" for gross motor development at 6 months. In contrast, the aRRs for all birth months, except for those for February and December, were larger than 1.0 ($p<0.05$), indicating a "greater risk" at 12 months of age. The aRRs for latitude also remained significant after adjustment, with latitudes of 35°N or greater and less than 40°N resulting in the highest aRRs.

The tendency of "the risk being highest for the summer-born and least for the winter-born" was also observed when the participants were divided into the four groups differing in their trajectories of gross motor development (Table 3). The birth months of February, March, and January showed the highest percentages in the BEST group (82.5%, 81.4%, and 81.3%, respectively) and the lowest percentages in the WORST group (3.7%, 3.9%, and 4.4%, respectively). In contrast, the birth months of July, August, and June showed the highest percentages in the WORST group (10.9%, 10.3%, and 9.4%, respectively), while those of August, September, and July showed the lowest percentages in the BEST group (62.2%, 63.8%, and 64.5%, respectively).

The birth months in the IMPROVED and WORSENED groups also showed a peak and a trough, although their months of occurrence differed greatly. On the one hand, in the IMPROVED group, the September-, August-, and October-born improved the most (22.7%, 21.8%, and 19.5%, respectively), whereas the March-, February -, and April-born improved the least (5.7%, 6.0%, and 6.0% respectively). On the other, in the WORSENED group, the April-,

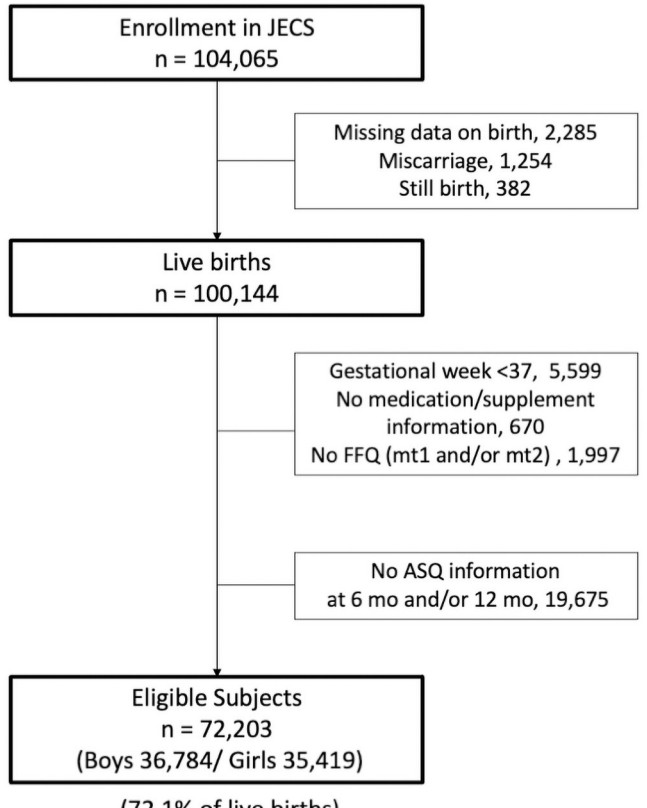

**Fig 1. Flowchart showing the number of children found eligible for this study among the participants in the JECS.**

May-, and June-born showed the greatest declines (10.2%, 9.4%, 9.0%, respectively), whereas the November-, October-, and September-born showed the least declines (5.2%, 5.3%, and 5.6%, respectively) (Table 3).

When the participants were stratified by latitude, those who resided 25˚N or above and less than 30˚N (Okinawa) performed the best, followed by those in areas 40˚N or above, 30˚N or above and less than 35˚N, and 35˚N or above and less than 40˚N (Table 3). Given that the number of participants in Okinawa was relatively small (n = 592), each of the other latitude groups was designated as the reference group. Nonetheless, the order of aRR values described above remained unchanged.

Finally, the study included 35,419 girls and 36,784 boys. Girls significantly underperformed at both time points ($p<0.001$): 7,590 girls (21.4%) and 7,370 boys (20.0%) were below the cut-off at 6 months, and the corresponding numbers at 12 months were 5,223 girls (14.8%) and 5,037 boys (13.7%). Of those who failed at both time points, 2,669 were girls (7.5%) and 2,437 were boys (6.6%). The aRR of the birth month for gross motor development stratified by gender showed a trend similar to that of the total aRR: the aRR for girls was higher than that for boys overall except for the birth month of February (Fig 2).

## Discussion

Gross motor development at 6 and 12 months of age was strongly associated with birth month in one of the world's largest national cohort studies. The one-year cyclical pattern of "higher

**Table 1. Participant characteristics.**

| Independent variable | | Total n = 72,203 | ASQ at 6 mo Gross Motor score <22.25 n = 14,960 (20.7%) | p value | ASQ at 12 mo Gross Motor score <21.49 n = 10,260 (14.2%) | p value |
|---|---|---|---|---|---|---|
| Birth month | | | | <0.001 | | <0.001 |
| January | | 5,594 | 704 (12.6) | | 592 (10.6) | |
| February | | 5,109 | 491 (9.6) | | 591 (11.6) | |
| March | | 5,571 | 537 (9.6) | | 715 (12.8) | |
| April | | 5,539 | 591 (10.7) | | 826 (14.9) | |
| May | | 5,781 | 896 (15.5) | | 949 (16.4) | |
| June | | 5,518 | 1,316 (23.9) | | 1,017 (18.4) | |
| July | | 6,408 | 1,799 (28.1) | | 1,178 (18.4) | |
| August | | 7,275 | 2,329 (32.0) | | 1,169 (16.1) | |
| September | | 7,467 | 2,322 (31.1) | | 1,012 (13.6) | |
| October | | 6,849 | 1,869 (27.3) | | 895 (13.1) | |
| November | | 5,622 | 1,146 (20.4) | | 684 (12.2) | |
| December | | 5,470 | 960 (17.6) | | 632 (11.6) | |
| (Region) | (Latitude) | | | <0.001 | | <0.001 |
| Hokkaido | 43.1 | 5,907 | 906 (15.3) | | 720 (12.2) | |
| Miyagi | 38.3 | 6,250 | 1,426 (22.8) | | 912 (14.6) | |
| Fukushima | 37.7 | 9,874 | 2,279 (23.1) | | 1,276 (12.9) | |
| Toyama | 36.7 | 4,210 | 957 (22.7) | | 743 (17.7) | |
| Shinshu | 36.3 | 2,084 | 564 (27.1) | | 367 (17.6) | |
| Yamanashi | 35.7 | 3,130 | 688 (22.0) | | 489 (15.6) | |
| Chiba | 35.6 | 3,695 | 813 (22.0) | | 534 (14.5) | |
| Tottori | 35.5 | 2,362 | 543 (23.0) | | 378 (16.0) | |
| Kanagawa | 35.3 | 4,785 | 874 (18.3) | | 656 (13.7) | |
| Aichi | 35.2 | 3,765 | 754 (20.0) | | 590 (15.7) | |
| Kyoto | 35.0 | 2,385 | 535 (22.4) | | 377 (15.8) | |
| Hyogo | 34.7 | 3,553 | 639 (18.0) | | 489 (13.8) | |
| Osaka | 34.7 | 5,729 | 1,120 (19.6) | | 797 (13.9) | |
| Fukuoka | 33.6 | 5,551 | 1,083 (19.5) | | 743 (13.4) | |
| Kochi | 33.5 | 4,830 | 987 (20.4) | | 676 (14.0) | |
| Kumamoto | 32.8 | 2,122 | 470 (22.2) | | 291 (13.7) | |
| Miyazaki | 31.8 | 1,379 | 257 (18.6) | | 183 (13.3) | |
| Okinawa | 26.3 | 592 | 65 (11.0) | | 39 (6.6) | |

risk for the summer-born and lower risk for the winter-born" was observed both at 6 and 12 months of age, with a much wider gap between the peak and the trough of the aRR at 6 months old, and with the peak shifted slightly earlier at 12 months. These findings could indicate potential "disadvantages in the winter" and "advantages in the summer" that initially affect brain development during early pregnancy—the most crucial period for development of the central nervous system—and which cyclically and continuously influence brain growth throughout the first year of life.

Although a limited number of studies have investigated the association between birth month and early gross motor development, the results of this study are in line with previous smaller studies in Japan, the U.S.A., and China. A Japanese study (n = 742 infants) found that March/April-born performed the best, although the seasonality trend disappeared at 14 months of age [12]. Among infants in the U.S.A. (Denver), summer/fall-born started crawling

**Table 2. Gross motor development at 6 and 12 months by birth month and latitude.**

| | Unadjusted MODEL Failed at 6 mo | | | Adjusted MODEL Failed at 6 mo* | | | Unadjusted MODEL Failed at 12 mo | | | Adjusted MODEL Failed at 12 mo* | | |
|---|---|---|---|---|---|---|---|---|---|---|---|---|
| | RR | [95% CI] | p | aRR | [95% CI] | p | RR | [95% CI] | p | aRR | [95% CI] | p |
| **Birth month** | | | | | | | | | | | | |
| January | Reference | | | Reference | | | Reference | | | Reference | | |
| February | 0.77 | (0.68 0.86) | <0.001 | 0.75 | (0.65 0.86) | <0.001 | 1.09 | (0.98 1.23) | 0.119 | 1.12 | (0.98 1.29) | 0.102 |
| March | 0.77 | (0.69 0.86) | <0.001 | 0.76 | (0.67 0.88) | <0.001 | 1.21 | (1.09 1.35) | <0.001 | 1.21 | (1.05 1.38) | 0.007 |
| April | 0.85 | (0.76 0.95) | 0.003 | 0.82 | (0.72 0.94) | 0.003 | 1.41 | (1.27 1.57) | <0.001 | 1.48 | (1.30 1.68) | <0.001 |
| May | 1.23 | (1.11 1.36) | <0.001 | 1.21 | (1.07 1.36) | 0.002 | 1.55 | (1.40 1.72) | <0.001 | 1.63 | (1.43 1.84) | <0.001 |
| June | 1.90 | (1.73 2.08) | <0.001 | 1.84 | (1.65 2.06) | <0.001 | 1.74 | (1.58 1.93) | <0.001 | 1.84 | (1.63 2.09) | <0.001 |
| July | 2.23 | (2.04 2.43) | <0.001 | 2.15 | (1.93 2.39) | <0.001 | 1.74 | (1.57 1.92) | <0.001 | 1.79 | (1.58 2.02) | <0.001 |
| August | 2.54 | (2.33 2.76) | <0.001 | 2.51 | (2.27 2.78) | <0.001 | 1.52 | (1.37 1.67) | <0.001 | 1.57 | (1.39 1.77) | <0.001 |
| September | 2.47 | (2.27 2.69) | <0.001 | 2.43 | (2.20 2.70) | <0.001 | 1.28 | (1.16 1.42) | <0.001 | 1.34 | (1.18 1.52) | <0.001 |
| October | 2.18 | (2.00 2.38) | <0.001 | 2.12 | (1.91 2.35) | <0.001 | 1.24 | (1.12 1.37) | <0.001 | 1.24 | (1.09 1.41) | 0.001 |
| November | 1.63 | (1.48 1.79) | <0.001 | 1.61 | (1.44 1.80) | <0.001 | 1.15 | (1.03 1.29) | 0.012 | 1.18 | (1.03 1.35) | 0.018 |
| December | 1.40 | (1.27 1.54) | <0.001 | 1.35 | (1.20 1.52) | <0.001 | 1.09 | (0.98 1.22) | 0.117 | 1.11 | (0.97 1.28) | 0.128 |
| **Latitude** | | | | | | | | | | | | |
| ≥25˚N, <30˚N | Reference | | | Reference | | | Reference | | | Reference | | |
| ≥30˚N, <35˚N | 1.75 | (1.37 2.23) | <0.001 | 1.85 | (1.38 2.48) | <0.001 | 2.10 | (1.53 2.88) | <0.001 | 1.80 | (1.27 2.54) | 0.001 |
| ≥35˚N, <40˚N | 1.97 | (1.55 2.52) | <0.001 | 2.10 | (1.57 2.81) | <0.001 | 2.26 | (1.65 3.10) | <0.001 | 1.87 | (1.33 2.64) | <0.001 |
| ≥40˚N | 1.37 | (1.07 1.76) | 0.014 | 1.55 | (1.15 2.09) | 0.005 | 1.87 | (1.35 2.58) | 0.003 | 1.61 | (1.13 2.30) | 0.009 |

Total numbers for the unadjusted and adjusted models were 72,203 and 48,805, respectively.

*Adjusted Relative Risk (aRR): Multivariate analyses of birth month adjusted for latitude, time spent outside per day, pre-pregnancy BMI, maternal age, parity, assisted reproductive treatments, threatened abortion/premature labor, neuropsychotropic drugs, smallness for gestational age, multiple birth, breech/foot/other abnormal presentation, induced delivery, C-section, labor>24 h, Apgar score (5 min), umbilical cord/placenta problems, meconium staining, neonatal transportation, respiratory distress, hyperbilirubinemia, intrauterine growth restriction, maternal smoking and drinking during pregnancy, maternal education, and family income.

three weeks later than winter/spring-born (n = 425), while among infants in China, winter-born scored higher on cognitive and psychomotor development tests at 8 to 10 months of age (n = 650) [14,16]. McGrath et al. [8] reported that winter/spring-born showed superior motor development at 8 months and 4 years but not at 7 years of age among 22,123 participants. However, a study in Canada (n = 145) found no statistically significant seasonal differences at 7 months of age [15].

Our findings can be explained by scrutinizing the association between critical windows of brain development and seasonality. Development of the human brain starts soon after conception and continues into early adulthood [26]. The period from conception to the end of infancy is known as a critical window of developmental brain plasticity and growth [27]. In particular, the first trimester is believed to be crucial for the central nervous system [28]. In the present study, August-born infants underperformed in the gross motor domain at both time points. If a child was born in the summer (August), the critical window of brain development would have been in late autumn to winter (November to February). Likewise, the critical window of brain development for the February-born, who performed the best in this study, would have been in the summer (June to September). The finding that the highest proportion in the "BEST" group was winter-born, and the highest proportion in the "WORST" group was summer-born, also indicates that more risk factors may exist during the winter and more protective factors may be present in the summer.

**Table 3. Trajectory of gross motor at 6 & 12 months by birth month/latitude.**

| | ≧cutoff at 6 mo & 12 mo (BEST) | <cutoff at 6 mo ≧cutoff at 12 mo (IMPROVED) | ≧cutoff at 6 mo <cutoff at 12 mo (WORSENED) | <cutoff at 6 mo & 12 mo (WORST) | TOTAL |
|---|---|---|---|---|---|
| **Birth Month** | | | | | |
| January | 4,546 (81.3) | 456 (8.2) | 344 (6.2) | 248 (4.4) | 5,594 |
| February | 4,214 (82.5) | 304 (6.0) | 404 (7.9) | 187 (3.7) | 5,109 |
| March | 4,537 (81.4) | 319 (5.7) | 497 (8.9) | 218 (3.9) | 5,571 |
| April | 4,381 (79.1) | 332 (6.0) | 567 (10.2) | 259 (4.7) | 5,539 |
| May | 4,341 (75.1) | 491 (8.5) | 544 (9.4) | 405 (7.0) | 5,781 |
| June | 3,703 (67.1) | 798 (14.5) | 499 (9.0) | 518 (9.4) | 5,518 |
| July | 4,132 (64.5) | 1098 (17.1) | 477 (7.4) | 701 (10.9) | 6,408 |
| August | 4,523 (62.2) | 1,583 (21.8) | 423 (5.8) | 746 (10.3) | 7,275 |
| September | 4,760 (63.8) | 1,695 (22.7) | 385 (5.2) | 627 (8.4) | 7,467 |
| October | 4,617 (67.4) | 1,337 (19.5) | 363 (5.3) | 532 (7.8) | 6,849 |
| November | 4,159 (74.0) | 779 (13.9) | 317 (5.6) | 367 (6.5) | 5,622 |
| December | 4,176 (76.3) | 662 (12.1) | 334 (6.1) | 298 (5.5) | 5,470 |
| **Latitude** | | | | | |
| ≥25˚N, <30˚N | 504 (85.1) | 49 (8.3) | 23 (3.9) | 16 (2.7) | 592 |
| ≥30˚N, <35˚N | 16,994 (73.4) | 2,991 (12.9) | 1,614 (7.0) | 1,565 (6.8) | 23,164 |
| ≥35˚N, <40˚N | 29,986 (70.5) | 6,232 (14.7) | 3,121 (7.3) | 3,201 (7.5) | 42,540 |
| ≥40˚N | 4,605 (78.0) | 582 (9.9) | 396 (6.7) | 324 (5.5) | 5,907 |
| Total | 52,089 (72.1) | 9,854 (13.7) | 5,154 (7.1) | 5,106 (7.1) | 72,203 |

This "risk in winter and benefit in summer" effect could possibly affect children not only during their prenatal period but also during the first 12 months of life. The finding that the highest proportions of infants in the "IMPROVED" group were August- and September-born and those in the "WORSENED" group were April- and May-born also suggests a negative effect of winter and positive effect of summer on brain development even after birth, because spring-born infants experience winter just before their first birthday whereas summer-born infants experience summer just before theirs. Bai at el. [16] pointed out the possibility of positive climatic factors during summer, which provide a more preferable environment for infants to explore around, as well as more nutritious food when they start weaning. The narrowing of the gap between the peak and the trough of the cyclical aRR trends at 12 months of age may also indicate that prenatal negative effects on summer-born infants (particularly during the first trimester) gradually subside, or that prenatal positive effects on winter-born infants gradually diminish due to the environment after birth. In other words, the results likely indicate the importance of seasonal factors among other environmental factors, as well as the plasticity of brain in the first 12 months after birth.

Birth month is a proxy for various intermingled biometeorological factors, such as sunlight, temperature, humidity, physical activity, nutritional intake, infection, and metabolic/endocrinological status. As most of the vitamin D we obtain requires being exposed to the sun, vitamin D insufficiency/deficiency could be a possible major risk factor in the winter. Vitamin D is crucial not only from an osteological but also a neurodevelopmental point of view, which includes gross motor development [29–32]. Clear seasonality of 25-HydrovitaminD (25OHD) levels has been found in previous studies in Japan, including a JECS Pilot Study on 126 children (2 to 4 years old) [33,34]. However, in that study, no significant latitude-dependent

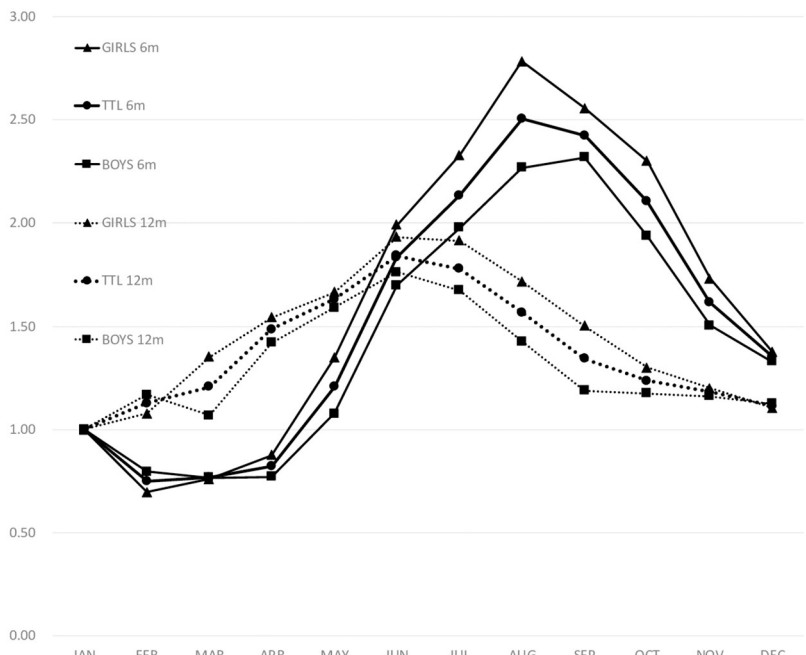

**Fig 2. Adjusted Relative Risk (aRR) for possible gross motor delay as a function of birth month and gender.**
Reference: January. 6 months: Solid line, 12 months: Dotted line. TTL: Total participants. For outcomes at 6 months, all months except April for girls and May for boys were statistically significant. For outcomes at 12 months, the months between March and November overall, between March and October for girls, and between April and August for boys were statistically significant (p<0.05).

differences were found among three regions (32˚N–36˚N), most likely because the latitude gap was narrower than in the present studies, the biometeorological factors involved were more complex, or both.

A positive association between Vitamin D sufficiency and motor development has been found in previous studies [32,35,36]. Tavakolizadeh et al. found a strong correlation between sufficient vitamin D and gross motor development among 12-month-old children [32]. At 6 months of age, summer-born infants are likely to be doubly deficient in vitamin D, from both the ongoing insufficiency/deficiency when they turned 6 months in the winter and from the prenatal vitamin D deficiency in the first trimester. Together with the result that their gross motor development improved the most at 12 months of age in the summer, the present results may indicate the importance of sufficient vitamin D intake from conception to early childhood. This implication of the present study—that vitamin D deficiency is a putative seasonal risk factor—could be clinically very relevant in Japan, where people are generally over-cautious about sun exposure, vitamin D-fortified food is extremely limited, and diagnosed cases of vitamin D-deficiency and rickets are on the rise [37].

Another possible seasonal risk factor characteristic of winter is influenza virus infection. Offspring exposed to influenza during between 0 and 8 gestational weeks showed a slight delay in their psychomotor development at 6 months in the Norwegian Influenza Pregnancy Cohort Study [38]. In addition, although the findings are inconclusive, some studies suggest that prenatal influenza virus infection may increase the risk of ASD among offspring, and that gross motor delay could be displayed as an early marker of later-diagnosed ASD [39].

Interestingly, among the five domains of the ASQ-3, only the gross motor domain showed clear seasonality and outperformance by male infants at both time points. Given the fact that

boys generally outnumber girls in prevalence of neurodevelopmental disorders, such as ASD, ADHD, intellectual disability, and language delay, [40,41], and a growing number of studies have reported that being winter/spring-born is a risk factor for ASD [3,9,42,43], and the fact that various combinations between environmental and genetic factors contribute to the etiology of neurodevelopmental disorders, it is likely that a delay in gross motor development at this age could be outgrown by most infants. Further follow-up is necessary to confirm the implication of infant gross motor problems, including a possibility of an early sign for later-diagnosed neurodevelopmental disorders of small subgroups.

Seasonality includes not only biometeorological but also social factors, particularly for older children. Age at school entrance is considered to affect the youngest children in class [44]. As all schools in Japan start in April, children born just before the cut-off (in March) are considered to be disadvantaged compared to April-born students. Kawaguchi [45] concluded that both males and females born between April and June have a higher educational level than those born between January and March, and for males, even a higher income in the long run. Given that winter-born infants outperformed others in gross motor development up until 12 months of age in this study, the ongoing JECS cohort needs to be carefully followed up with various seasonal perspectives on the one hand, and a wider range of future developmental issues on the other.

Although boys and girls showed similar seasonal trends, boys outperformed girls, and the discrepancy between the peak and the trough of the aRR among boys was narrower than that for girls, indicating that girls are more susceptible to seasonality with regard to gross motor development. In other words, winter-born girls benefited more and summer-born girls were handicapped more than boys. As girls outperformed boys in all of the other four skill domains —communication, fine motor, problem solving, and personal-social—a more thorough and long-term investigation on this ongoing cohort is necessary.

The strengths of the study lie in the data, in which a nationally representative dataset of large sample size was merged with the meteorological agency data. The JECS dataset covered a wide range of the country—from subarctic (Hokkaido) to subtropical (Okinawa) but mostly temperate zone—with information on various obstetric risk factors that allowed us to adjust the model. Furthermore, birth month information was collected accurately from medical record transcripts. Finally, motor development was assessed by the ASQ-3, an internationally standardized screening tool for assessing child development.

One limitation of the study is that because individual biological data to assess the impacts of seasonal factors, such as infections (i.e., influenza, cytomegalovirus, rubella) and serum vitamin D (25(OH)D), were not included, the mechanism we proposed to explain the strong association between birth month and gross motor development at 6 and 12 months of age is only speculative. Another limitation is that, although internationally validated and utilized, the ASQ-3 was conducted not by trained professionals but by guardians. Finally, although gross motor development is well recognized as an appropriate developmental indicator, particularly in early childhood, other developmental domains were not included.

## Conclusion

Among the JECS participants, birth month was strongly associated with gross motor development at 6 months, with summer-born infants underperforming and winter-born infants over-performing. Although this difference decreased as they grew, the tendency still remained at 12 months of age. The study results indicate positive biometeorological effects of summer and negative effects of winter during not only the prenatal but also the postnatal period. While further follow-up on this JECS cohort will confirm the long-term influence of birth month on

child neuromotor development, an important clinical implication of the present study is that seasonality during the prenatal and neonatal periods needs to be incorporated in clinical research to properly assess the lifetime health of offspring.

## Supporting information

**S1 Table. Participant characteristics.**
(DOCX)

## Acknowledgments

The authors are grateful to all of the JECS participants; all individuals involved in data collection; the Japan Environment and Children's Study Group (See Appendix); Nagamasa Maeda, Mikiya Fujieda, Naomi Mitsuda, Atsuko Mori, and Naw Awn J-P of the Kochi Regional Centre of the JECS; and Sifa Marie Joelle Muchanga of the National Center for Global Health and Medicine. The findings and conclusions of this article are solely the responsibility of the authors and do not represent the official views of the Ministry of the Environment, Japan.

Members of the JECS Group as of 2021: Michihiro Kamijima (principal investigator, Nagoya City University, Nagoya, Japan), Shin Yamazaki (National Institute for Environmental Studies, Tsukuba, Japan), Yukihiro Ohya (National Center for Child Health and Development, Tokyo, Japan), Reiko Kishi (Hokkaido University, Sapporo, Japan), Nobuo Yaegashi (Tohoku University, Sendai, Japan), Koichi Hashimoto (Fukushima Medical University, Fukushima, Japan), Chisato Mori (Chiba University, Chiba, Japan), Shuichi Ito (Yokohama City University, Yokohama, Japan), Zentaro Yamagata (University of Yamanashi, Chuo, Japan), Hidekuni Inadera (University of Toyama, Toyama, Japan), Takeo Nakayama (Kyoto University, Kyoto, Japan), Hiroyasu Iso (Osaka University, Suita, Japan), Masayuki Shima (Hyogo College of Medicine, Nishinomiya, Japan), Youichi Kurozawa (Tottori University, Yonago, Japan), Narufumi Suganuma (Kochi University, Nankoku, Japan), Koichi Kusuhara (University of Occupational and Environmental Health, Kitakyushu, Japan), and Takahiko Katoh (Kumamoto University, Kumamoto, Japan).

## Author Contributions

**Conceptualization:** Kahoko Yasumitsu-Lovell.

**Data curation:** Kahoko Yasumitsu-Lovell, Masamitsu Eitoku, Narufumi Suganuma.

**Formal analysis:** Kahoko Yasumitsu-Lovell, Lucy Thompson.

**Funding acquisition:** Narufumi Suganuma.

**Investigation:** Kahoko Yasumitsu-Lovell, Christopher Gillberg.

**Methodology:** Kahoko Yasumitsu-Lovell, Masamitsu Eitoku, Narufumi Suganuma.

**Project administration:** Kahoko Yasumitsu-Lovell, Masamitsu Eitoku, Narufumi Suganuma.

**Resources:** Elisabeth Fernell, Narufumi Suganuma.

**Supervision:** Lucy Thompson, Elisabeth Fernell, Masamitsu Eitoku, Narufumi Suganuma, Christopher Gillberg.

**Validation:** Kahoko Yasumitsu-Lovell, Narufumi Suganuma.

**Visualization:** Kahoko Yasumitsu-Lovell.

**Writing – original draft:** Kahoko Yasumitsu-Lovell.

**Writing – review & editing:** Kahoko Yasumitsu-Lovell, Lucy Thompson, Elisabeth Fernell, Masamitsu Eitoku, Narufumi Suganuma, Christopher Gillberg.

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
