## [Decision Letter · Decision Letter 0]

18 Jan 2021

PONE-D-20-31154

Birth month and infant gross motor development – Results from the Japan Environment and Children’s Study (JECS)

PLOS ONE

Dear Dr. Suganuma,

Thank you for submitting your manuscript to PLOS ONE. After careful consideration, we feel that it has merit but does not fully meet PLOS ONE’s publication criteria as it currently stands. Therefore, we invite you to submit a revised version of the manuscript that addresses the points raised during the review process.

We look forward to receiving your revised manuscript.

Kind regards,

Maria Christine Magnus, MPH

Academic Editor

PLOS ONE

Additional Editor Comments:

We have received comments on your manuscript from two reviewers. I also have additional comments on your paper. Please address all comments carefully in your revised manuscript.

Comments:

1) Please clearly justify the cut-off values you have used for the neurodevelopmental scores.

2) You need to state how you dealt with missing information. Specifically, I assume that there was missing information in at least some of the covariates you adjusted for in the multivariable analysis. If there was no missing in the covariates, then this should be stated. If there was missing information, then you should consider multiple imputation.

3) Some of the covariates you adjusted for are likely to be mediators (potentially part of the explanatory pathway) and not confounders. I am for example thinking of all the pregnancy outcomes which you adjust for. You should clearly classify which covariates are confounders as opposed to potential mediators. You should also not adjust your main model for the potential mediators but include these in a second multivariable model.

4) I wonder why you chose to look at season of delivery and not season of conception? Since you mention that intra uterine vitamin D levels during the fist trimester might be part of the explanatory mechanism, I wonder if season of conception is not of more interest? Please carefully think this through and justify your decision.

5) Move all results from the methods to the results section. There should be no results tables or references to these in the methods.

6) The manuscipt would greatly benefit from language editing. I strongly suggest you consider sending the mansucript for professtional language editing.

7) Please clarify why you chose to use poission regression (IRR) instead of a log-binomial regression (RR)?

Journal Requirements:

2.) Thank you for including your ethics statement:  "The JECS protocol was reviewed and approved by the Ministry of the Environment’s Institutional Review Board on Epidemiological Studies and by the Ethics Committees of all participating institutions. Written informed consent was obtained from all

participating women and men. The below link contains the JECS ethical information (p17)

. " ext-link-type="uri" xlink:type="simple">http://www.nies.go.jp/kanko/news/38/38-4/jqjm1000000i2db3-att/jqjm1000000i2dc0.pdf".   

3.) In statistical methods, please clarify whether you corrected for multiple comparisons.

4.) Thank you for stating the following in the Acknowledgments Section of your manuscript:

'The Japan Environment and Children’s Study was funded by the Ministry of Environment, Japan.'

'The authors received no specific funding for this work.'

5.) One of the noted authors is a group or consortium Japan Environment and Children’s Study Group. In addition to naming the author group, please list the individual authors and affiliations within this group in the acknowledgments section of your manuscript. Please also indicate clearly a lead author for this group along with a contact email address.

Reviewers' comments:

Reviewer's Responses to Questions

**Comments to the Author**

1. Is the manuscript technically sound, and do the data support the conclusions?

Reviewer #1: Yes

Reviewer #2: Yes

2. Has the statistical analysis been performed appropriately and rigorously? 

Reviewer #1: Yes

Reviewer #2: Yes

3. Have the authors made all data underlying the findings in their manuscript fully available?

Reviewer #1: Yes

Reviewer #2: Yes

4. Is the manuscript presented in an intelligible fashion and written in standard English?

Reviewer #1: Yes

Reviewer #2: Yes

5. Review Comments to the Author

Reviewer #1: Abstract:

-It is appropriate to mentioned P values in significant result.

Methods:

-The following phrase, Fig 1, Table 1, 2 and 3 which is given in the statistical analysis section, should be given in the results section.

“The means of maternal age at enrolment for the study subjects and 81 the excluded were 30.8 (95% CI: 30.0, 30.9) and 30.3 (95% CI: 30.2, 30.3) years old”

Discussion:

In methods section mentioned:

-“The questionnaires covered various topics, including lifestyle, socio-economic status (SES), diet, medical history, and medication”.

And in Table 1, many independent variables were mentioned in the study cases, such as prepregnancy BMI, hyperbilirubinemia, Apgar score, C-section, which there were significant difference. But in the discussion, there is no mention of the possibility of the effect of the independent variables.

Some abbreviations need to be mentioned in full wherever they are first mentioned: EPDS, MR, LD.

Figure 2 is not clear.

Reviewer #2: Introduction.

The study needs more information on previous findings on relationship between birth month and gross motor skills in early childhood, even the previous findings were not from large cohort samples. What has the literature found? Has the literature found significantly positive/negative relationship between the two variables of interest? Has any literature investigated the trajectories of motor development of young children? I think it needs to add more motivations of the study in this section.

Methods.

1. What’s the strategies of the sampling? What’s the process of data collection? Such as data on vitamin D intake, etc. Readers may want to know more detailed information on sampling and data collection.

2. As the measurement of child gross motor development, how about the internal consistency of the ASQ-3 for the sample used in the study? I would suggest the authors reporting this in the methods section.

Results.

1. Table 1 looks too big. And it was inserted in the wrong place of the manuscript. It is also true for Tables 2 3.

2. On line 173, a quotation mark has been missed.

Discussion.

1. Again, more literature should be revisited in this section regarding to the issues that are discussed here.

2. What are the implications of the study? The implications should be discussed here or in the conclusion section.

6. PLOS authors have the option to publish the peer review history of their article (what does this mean?). If published, this will include your full peer review and any attached files.

Reviewer #1: No

Reviewer #2: No

---

## [Author Response · Author response to Decision Letter 0]

14 Apr 2021

Thank you for letting us submit the revised manuscript. Please be informed that all our responses are stated in the cover letter and our responses to reviewers. As it contains two tables, I cannot copy/paste in this section, so it would be highly appreciated if you could read the file itself.

---

## [Decision Letter · Decision Letter 1]

29 Apr 2021

Birth month and infant gross motor development: Results from the Japan Environment and Children’s Study (JECS)

PONE-D-20-31154R1

Dear Dr. Suganuma,

We’re pleased to inform you that your manuscript has been judged scientifically suitable for publication and will be formally accepted for publication once it meets all outstanding technical requirements.

Kind regards,

Maria Christine Magnus, MPH

Academic Editor

PLOS ONE

Additional Editor Comments (optional):

Reviewers' comments:

Reviewer's Responses to Questions

**Comments to the Author**

1. If the authors have adequately addressed your comments raised in a previous round of review and you feel that this manuscript is now acceptable for publication, you may indicate that here to bypass the “Comments to the Author” section, enter your conflict of interest statement in the “Confidential to Editor” section, and submit your "Accept" recommendation.

Reviewer #1: All comments have been addressed

Reviewer #2: All comments have been addressed

2. Is the manuscript technically sound, and do the data support the conclusions?

Reviewer #1: Yes

Reviewer #2: Yes

3. Has the statistical analysis been performed appropriately and rigorously? 

Reviewer #1: Yes

Reviewer #2: Yes

4. Have the authors made all data underlying the findings in their manuscript fully available?

Reviewer #1: Yes

Reviewer #2: Yes

5. Is the manuscript presented in an intelligible fashion and written in standard English?

Reviewer #1: Yes

Reviewer #2: Yes

6. Review Comments to the Author

Reviewer #1: The revised manuscript is acceptable. The authors have adequately addressed my comments raised in a previous round of review and I feel that this manuscript is now acceptable for publication.

Reviewer #2: All my comments have been addressed. The format and the language of the manuscript now are suitable for publication.

7. PLOS authors have the option to publish the peer review history of their article (what does this mean?). If published, this will include your full peer review and any attached files.

Reviewer #1: No

Reviewer #2: No

---

## [Editor Report · Acceptance letter]

3 May 2021

PONE-D-20-31154R1 

Birth month and infant gross motor development: Results from the Japan Environment and Children’s Study (JECS) 

Dear Dr. Suganuma:

I'm pleased to inform you that your manuscript has been deemed suitable for publication in PLOS ONE. Congratulations! Your manuscript is now with our production department. 

Kind regards, 

on behalf of

Dr. Maria Christine Magnus 

Academic Editor

PLOS ONE